# Electrodialytic Desalination of Tobacco Sheet Extract: Membrane Fouling Mechanism and Mitigation Strategies

**DOI:** 10.3390/membranes10090245

**Published:** 2020-09-21

**Authors:** Shaolin Ge, Zhao Zhang, Haiyang Yan, Muhammad Irfan, Yingbo Xu, Wei Li, Huangying Wang, Yaoming Wang

**Affiliations:** 1China Tobacco Anhui Industrial Co., LTD, Hefei 230088, China; slge@mail.ustc.edu.cn (S.G.); zhaoz@mail.ustc.edu.cn (Z.Z.); xuybah@hotmail.com (Y.X.); 2CAS Key Laboratory of Soft Matter Chemistry, Laboratory of Functional Membranes, School of Chemistry and Materials Science, University of Science and Technology of China, Hefei 230026, China; oceanyan@ustc.edu.cn (H.Y.); engr_muhammad.irfan@hotmail.com (M.I.); why921@mail.ustc.edu.cn (H.W.); 3Hefei ChemJoy Polymer Materials, Co., LTD, Hefei 230601, China; liwei8991@126.com

**Keywords:** ion exchange membrane, electrodialysis, desalination, fouling, separation

## Abstract

In the papermaking industry (reconstituted tobacco), a large number of tobacco stems, dust, and fines are discharged in the wastewater. This high salinity wastewater rich in ionic constituents and nicotine is difficult to be degraded by conventional biological treatment and is a serious threat that needs to be overcome. Electrodialysis (ED) has proved a feasible technique to remove the inorganic components in the papermaking wastewater. However, the fouling in ion exchange membranes causes deterioration of membranes, which causes a decrease in the flux and an increase in the electrical resistance of the membranes. In this study, the fouling potential of the membranes was analyzed by comparing the properties of the pristine and fouled ion exchange membranes. The physical and chemical properties of the ion exchange membranes were investigated in terms of electrical resistance, water content, and ion exchange capacity, as well as studied by infrared spectroscopy (IR) spectra, scanning electron microscopy (SEM), and energy dispersive spectroscopy (EDS) analyses. The results indicated that the membrane fouling is caused by two different mechanisms. For the anion exchange membranes, the fouling is mainly caused by the charged organic anions. For the cation exchange membrane, the fouling is caused by minerals such as Ca^2+^ and Mg^2+^. These metal ions reacted with OH^−^ ions generated by water dissociation and precipitated on the membrane surface. The chemical cleaning with alkaline and acid could mitigate the fouling potential of the ion exchange membranes.

## 1. Introduction

In the tobacco industry, a large number of tobacco stems, dust, and fines are produced as waste, which occurs in almost one-third of the raw materials [1]. Papermaking reconstituted tobacco (PRT) is an emerging technology to recycle tobacco wastes, reduce environmental pollution, and save cultivated lands. The chemical, physiological, and biological properties of cigarettes made from papermaking tobacco sheet can be manually manipulated with the addition of external chemicals and additives including sugars and flavors. Tobacco made from reconstituted tobacco has several superiorities in structural strength, combustion performance, and tar delivery compared to natural tobacco [2]. The lower chemical contents and faster combustion of PRT than natural tobacco can decrease the inhaled nicotine and can also reduce the puff number of cigarettes [3]. It is reported that the PRT constitutes 20–25% of the tobacco raw materials for several high-grade cigarettes such as Marlboro, Kent, Winston, and Camel [4]. Compared to natural tobacco leaf, the papermaking tobacco sheet is rich in inorganic constituents like K^+^, Cl^−^, SO_4_^2−^, and NO^3−^. Numerous studies have shown that the ionic components have significant influences on the sensory taste of the cigars and the thermal behavior in cigarette smoke [5,6]. Generally, chloride and nitrate/nitrite ions are considered as undesirable components which affect the moisture absorptivity and combustibility of cigarette as well as harmful carbonyl compounds delivery in the mainstream smoke [7,8]. According to the PRT standards for cigarettes in China, the chloride and nitrate concentrations in the PRT should be lower than 0.8 wt.% and 0.5 wt.%, respectively. In contrast, the divalent or multivalent alkaline metals are usually considered as desirable components which have catalytic roles in the thermal degradation and char formation of biomass [9]. Therefore, it is important to manually manipulate the ionic components of the tobacco sheet extract.

To selectively remove the undesired inorganic matters in the tobacco extract, several separation technologies including multistage continuous countercurrent extraction [10], integration of solvent extraction, condensation and frozen centrifugation [11], microorganism fermentation [12], and ion exchange [13] and absorption [14] were implemented. Generally, these techniques exhibited unsatisfying performances on selectivity removal of Cl^−^ and NO^3−^ ions and also have drawbacks such as large consumption of chemicals, high operating cost, and very easy to cause secondary pollution. Contrary to the separation technologies, electrodialysis (ED) does not suffer from the major drawbacks such as the use of costly chemicals, generation of large amounts of wastes, and short lifetimes of absorbents. Therefore, ED is considered an environmentally friendly technology, which has found a multitude of applications in water desalination, cleaning production or separation, resources recycling, power generation, and sensitive electrode preparation [15,16,17]. Considering the tobacco industry, Bazinet and co-authors [18] used the ED for the electromigration of tobacco polyphenols, with an overall demineralization of 77%. In a subsequent study [19], the authors further improved the separation performances of tobacco polyphenols by extending the experimental time. Similarly, our previous studies have proven the feasibility of ED for the removal of inorganic ions and decrease the harmful component delivery in cigarette smoke [20,21,22]. In these studies, ED was not only used for the removal of the harmful chloride and nitrate ions, but also for selectively adjusting the monovalent and divalent inorganic ion composition. During the combustion of cigarettes, the main harmful volatile carbonyl compounds such as CO, NH_3_, HCN, phenol, and crotonaldehyde are released from the pyrolysis of carbohydrate, including sugar, starch, and cellulose, etc. Our previous studies [21,22] have proven that ED is beneficial to selectively decrease the harmful carbonyl components in cigarette smoke. Based on the experimental results, a 10 tons/day demonstration project has been developed in China Tobacco Anhui Reconstituted Tobacco Technology Co., Ltd. (Bengbu, Anhui Province, China). The reconstituted tobacco sheet with ED treatment exhibited improved performances on the taste and delivery of harmful mainstream tobacco smoke compared to that of without the ED treatment. However, in the industrial application of ED on tobacco sheet extract, the stability of membranes is the foremost important thing to be considered. Membrane fouling causes the deterioration of ion exchange membranes, witnessed by a decrease in the flux and increase in the electrical resistance of the ED stack [23,24,25]. The compositions of tobacco extract are very complex, including thousand kinds of matters [26] such as minerals, sugars, organic acids, amino acids, carbohydrates, polyphenols, esters, alcohols, pigments, etc. In fact, there are numerous studies that investigated the fouling of membrane during the electrodialysis process. According to the review summarized by Bazinet [27], there are usually four kinds of fouling in the ion exchange membrane such as colloidal, organic, scaling, and biofouling. To elucidate the evolution of fouling in the ion exchange membranes, various methods have been used including visualization of the fouling by photo imaging, scanning electron microscopy (SEM) and atomic force microscopy (AFM), monitoring the evolution of the membrane characteristics such as membrane electrical resistance and conductivity, voltammetry and chronopotentiometry, transport number, zeta potential and contact angles, and analysis the fouling composition such as spectroscopy, chromatography, and biofouling characterization. To mitigate the fouling of the membranes, numerous strategies allowing the fouling prevention have been proposed such as by using a cleaning agent, modifying the ion exchange membranes, introducing an elaborated pretreatment, involving mechanical cleaning such as ultrasound, vibration, and air sparge, and by changing regimes of ED treatment. However, there are rare studies on the fouling of ion exchange membranes for treating tobacco sheet extract from a practical production line. Here, we developed readily adoptable methods to monitor the fouling of membranes, and elucidated the possible mechanisms for membrane fouling by tiny characterizations and by measuring the evolutions of the main properties of membranes, and finally proposed an effective method to clean the fouled membranes with improved performances. This kind of investigation of fouling in the ion exchange membranes can be extended to other kinds of plant extracts. Therefore, the main objectives of this research were to investigate the stability of the ED process for desalting of papermaking tobacco sheet extract, to enlighten the mechanism of fouling, and to alleviate the fouling on the ion exchange membranes.

## 2. Materials and Methods

### 2.1. Materials

Tobacco sheet extract (dissolved solid content ~7%) used in this experiment was supplied by China Tobacco Anhui Industrial Co., Ltd., Hefei, Anhui Province, China. This tobacco extract was pretreated by centrifugation at 3000 rpm for 5 min to remove the suspended solids before the ED process. The main characteristics of the extract are shown in Table 1. The membranes used for the electrodialysis (ED) experiments were CJ-MA-4 (Hefei ChemJoy Polymers Co., Ltd., Hefei, China) and CJ-MC-4 (Hefei ChemJoy Polymers Co., Ltd., Hefei, China). All the chemicals were of analytical grade.

### 2.2. ED Set-Up

A self-assemble and experimental scale ED stack was used for the experiment and the schematic diagram of the stack configuration is illustrated in Figure 1. The ED stack was consisting of four parts. The 1st part of ED stack is one pair of electrodes, which were made of iridium-tantalum coating with a thickness of 1.5mm. The electrode was fixed into a piece of organic glasses plate with a pre-setting rectangular notch to form an even surface of electrodes on the organic glasses plate. The current was applied on the electrodes using a direct current power supply (WYL1703, Hangzhou Siling Electrical Instrument Ltd., Hangzhou, China). The 2nd part of stack consists of the membranes with an effective area of 189 cm^2^ for each piece of membrane. The repeating unit of stack is ten. The 3rd part of ED stack consists of the spacers with each thickness of 0.75mm to separate two neighboring membranes. The 4th part of ED stack consists of three beakers that were circulated with three submersible pumps (AP1000, Zhongshan Zhenghua Electronics Co. Ltd., Zhongshan, China, flow rate of 22 L/h) to form three circulating loops. The anode and cathode compartment were connected together with a 500-mL Na_2_SO_4_ solution (0.3 mol/L) as the rinse electrolyte. The dilute chamber was fed with tobacco extract; the concentrate chamber was fed with deionized water.

### 2.3. Characterization of Ion Exchange Membranes

The fouling evolution of the ion exchange membranes was analyzed by comparing the membrane properties between the fresh and fouled membranes. To determine the ion exchange capacity (IEC, expressed in mmol/g) of the anion exchange membrane, the membrane was firstly washed with deionized water and dried at 60 °C for 24 h under vacuum before the mass was recorded. Then, the anion exchange membrane was immersed into a Na_2_SO_4_ (0.5 mol/L) solution for 8 h to convert the membrane into the SO_4_^2−^ form. Finally, the amount of Cl^-^ ions was titrated with an AgNO_3_ (0.1mol/L) aqueous solution using K_2_CrO_4_ as a colorimetric indicator. The IEC values were determined by the amount of AgNO_3_ consumed in the titration and the mass of the dry membrane. To determine the ion exchange capacity of a cation exchange membrane, the membrane in H^+^ form was equilibrated in 1 mol/L aqueous NaCl solution for 24 h to release the H^+^ ions, and then titrated with standardized base using phenolphthalein as an acid-base indicator.

The transport number was estimated by measurement of the membrane potential of a counter-ion in the ion exchange membrane using Ag/AgCl electrodes in 0.01 M and 0.05 M KCl solutions, respectively. The resistance of the ion exchange membranes was measured in the self-made stack by impedance spectroscopy (IS) using an Autolab PGSTAT 30 (Eco Chemie, Netherland) in a frequency range from 1–107 Hz with an oscillating voltage of 100 mV amplitude at room temperature [28]. The current-voltage curve was determined according to our previous study [29]. The water content (*Wu*) of the ion exchange membrane was determined by the following Equation (1),
(1)$$Wu\left(\%\right)=\frac{Ww-Wd}{Wd}\times 100\%$$
where *W_w_* is the wet weight of the membrane, and *W_d_* is the dry weight of the membrane. It should be noted that for the determination of wet weight, the sample was equilibrated in deionized water at first, and then the surface water from the membrane sample was removed. To measure the dry weight of the membrane, the sample was dried in an oven for 6 h at 100 ± 5 °C until a constant weight was obtained.

Fourier transform infrared spectroscopy of pristine and fouling membranes were measured by using the attenuated total reflectance (ATR) technique with FT-IR spectrometer (Vector 22, Bruker, Karlsruhe, Germany) having a spectral range of 4000–500 cm^−1^ and a resolution of 2 cm^−1^.

### 2.4. Desalination Rate

In the membrane cleaning process, the desalination rate (*R*) of the tobacco extract was calculated using Equation (2),
(2)$$R=\frac{\sigma_{t}}{\sigma_{0}}$$
where *σ_t_* and *σ*_0_ are the solution conductivity in the dilute chamber at time *t* and 0, respectively. It should be noticed that it is easy to judge the effective of different proposed cleaning methods by monitoring the changes in the desalination rate in consecutive batch operations.

## 3. Results

### 3.1. Determination of Limiting Current Density (LCD)

For a typical ED process, the limiting current density (LCD) is a key factor that determines the fouling of ion exchange membranes. The ion exchange membranes are considered as solid electrolytes, and the ion transport velocity in the ion exchange membranes are believed to be faster than that in the bulk solution. The ion transport velocity discrepancy between the membrane and bulk solution is accelerated with increasing current density. The LCD is the case when the salt ion concentration at the membrane-solution interface is zero. If the LCD is reached, the water dissociation will appear at the surface of ion exchange membranes. In that case, water will split into protons and hydroxide ions; the latter will react with Mg^2+^ and Ca^2+^ ions, causing the fouling of membranes. Therefore, the applied current density of ED shall be lower than the LCD to prevent the membrane fouling of membranes. However, in practical applications, LCD is difficult to be real-time monitored. The LCD is highly dependent on the salt ion concentration, boundary layer thickness, and transport number of ion exchange membrane. The LCD of the ED process can be estimated using Equation (3) [30],
(3)$$i_{lim}=\frac{F\cdot D\cdot C}{\left(\bar{t}-t\right)\cdot \delta }$$
where *F* is Faraday constant; *D* is the diffusion coefficient of salt in the solution; *C* is the salt concentration in the dilute compartment; $\bar{t}$ and *t* are the transport numbers of counterions in the membrane and the solution, respectively; *δ* is the thickness of the boundary layer. It is clearly shown that for the ED experiment, the LCD is not a constant value. The value of LCD decreases with the desalting of salts. To simulate the whole desalination process of tobacco extracts, the I-V curves of different contents of tobacco extract in the range of 0.5–10% are determined and depicted in Figure 2. It is clearly shown that the LCD decreases with a decrease in tobacco extract concentration. For instance, the LCD is 9.8 mA/cm^2^ when the feed tobacco content is 10%, while the LCD decreases to 3.3 mA/cm^2^ when the feed tobacco content is reduced to 0.5%. The concentration polarization is a common reason for the fouling of the ion exchange membrane.

### 3.2. Continuous Batch ED Operation

To evaluate the potential fouling of the membranes, four batch experiments were consecutively performed without any chemical or hydraulic washing. The feed tobacco extract concentration was 7% and the initial current density of ED stack was 15 mA/cm^2^. These experimental conditions were the same as those used previously. Figure 3 shows the dependences of conductivity in the tobacco sheet extract for four consecutive batch ED operations. It can be seen that the conductivities decreased with time elapse for these individual operations. This indicates that most of the ionic electrolytes in the tobacco extract were removed during the ED process. However, the experimental period is extended after several couples of operations. For the 1st run, one batch experimental period is 25 min; one batch experiment time increases to 35 min for the 2nd run; and the experimental time further increases to 40 min at the 3rd and 4th batch of experiments. And the desalting performance in ED also decreased after several couples of operations. The most possible reason for that is that the membrane fouling has occurred on the surface of the membranes. Since membrane fouling leads to increased membranes resistance and higher energy consumption, thus this decreased the membrane separation performances.

### 3.3. The Main Property Comparison between Pristine and Fouled Membranes

To identify the reasons for membrane fouling, the main properties of the pristine and fouled membranes were compared and the results are listed in Table 2. It is noticed that only the IEC and area resistance were determined. The ion exchange capacity is one of the most crucial parameters of the ion exchange membranes, which reveal the number of fixed charges per unit weight of the dry polymer. In Table 2, it is demonstrated that the IECs of the fouled cation exchange membranes decreased compared to that of the anion exchange membranes. The decease of ion exchange membranes is understood since the large amounts of metal cations in the tobacco extract can be easily combined with the negative charged functional groups of the cation exchange membranes. However, in the case of the anion exchange membranes, the IECs of the fouled anion exchange membranes are even higher compared with that of the pristine anion exchange membranes. The IECs of fouled AEM are nearly two times those of the pristine membranes. A possibly cause is the adsorption of organic matters such as humic acids, organic acids, and amino acids on the surface of the anion exchange membrane. This is because the anion exchange membrane used for the experiment has a semi-interpenetrating polymer network structure that using polyvinylidene fluoride (PVDF) as the backbone and poly chloromethyl styrene quarternarized ammonia as the functional groups. These negative charged foulants have electrostatic interactions of the quaternary ammonia fixed groups of the anion exchange membranes [31]. Additionally, there was π-π (stacking) interaction of aromatic rings of membranes and foulants such as humic acid [27]. In this way, these anionic foulants would deposit on the membrane surface, and thus causing the fouling of the anion exchange membranes.

The resistance of ion exchange membranes is one important parameter that directly determines the energy consumption of the electrodialysis process. It is shown in Table 2 that both the area resistance of the fouled anion and cation exchange membranes are increased compared with the pristine membranes. For the cation exchange membranes, the area resistance of fouled membranes increased 2–3 times that of the pristine membranes. However, for the anion exchange membranes, the area resistance of fouled membranes increased nearly 5 times that of the pristine membranes. The fouling of ion exchange membranes leads to the increase of membrane resistance, which are possibly related to two causes. One cause is due to the formation of surface fouling layer and the other cause is due to the deposition of foulants on the functional groups of ion exchange membranes [27].

The water uptake is also an important parameter for an ion exchange membrane, which is highly determined by the nature of the polymer matrix, nature and concentration of fixed ionic moiety, the type of mobile ions, the cross-linking density, and the homogeneity of the membrane [32,33]. From Table 2, it can be seen that there is only a slight decrease in water uptake in the fouled anion exchange membranes compared with fresh anion exchange membranes. The water uptake of an ion exchange membrane is high correlating with other properties of membranes such as ion exchange capacity, and transport number, etc. In the case of fouled cation exchange membrane, the decrease in ion exchange capacity is the main reason for the decrease of water uptake. However, for the fouled anion exchange membrane, the increase of ion exchange capacity has not resulted in an obvious increase in water uptake. This suggests that the fouling mechanism of cation exchange membranes may be different from the anion exchange membranes. A detailed fouling mechanism will be discussed in the following sections combined with the characterization results.

The transport number is also an important parameter that indicates the selectivity of membranes. The transport number can be determined by the ratio of transport of electric charges by the counter-ions to the total electrical current through the membranes. There are no obvious differences in transport number for the fresh and fouled cation exchange membranes. A slight decrease in transport number is observed for the fouled anion exchange membranes compared with the pristine membrane.

### 3.4. Characterization of the Fouled Membranes

There are thousands of matters in the tobacco extract including sugars, amino acids, nicotine, proteins, inorganic ions such as potassium, sodium, chloride, nitrate, and sulfate, etc. To investigate the fouling of the ion exchange membranes during the desalting of tobacco sheet extract, the FT-IR spectra and SEM-EDS were used to characterize the fouled membrane. The membranes used for the experiments have a semi-interpenetrating polymer network structure that use PVDF as the backbone and quaternary ammonia and sulfonic acids as the functional groups of the anion and cation exchange membrane, respectively. Figure 4 shows the spectra of the pristine and fouled anion and cation membranes. It can be seen that there is no obvious change observed for cation exchange membrane absorption spectra before and after the ED experiments. In contrast, a new doublet is observed at 3750 cm^−1^ for the fouled anion exchange membranes compared with the fresh membranes. This is possibly attributed to the stretching and vibration of hydroxyl ions. In the ED process, water molecules easily tend to be dissociated at the surface of anion exchange membranes when the current density has reached the limiting current density. It is known that most of the organic foulants are negatively charged, thus fouled anion exchange membranes due to deposition and/or adsorption. However, more detailed information cannot be provided only by FT-IR spectra.

Appendix A indicates the SEM-EDS images for both the anion and cation exchange membranes after fouling. As observed from these images, a large number of precipitates found on both of the dilute and concentrate sides of the membranes. The EDS analysis of the cation exchange membranes facing the concentrate chamber indicated the presence of C, F, S, Na, Mg, and Ca. In comparison, the EDS images of the cation exchange membranes face the dilute chamber indicated the presence of the same elements except for the Na element. The C, F, and S are the elements of the backbone and functional groups of the investigated cation exchange membranes. The cation exchange membranes used in the experiments was composed of the polyvinylidene fluoride as the main backbone and sulfuric acid as the function group. The existence of sodium, magnesium, and calcium elements mainly comes from the tobacco extract which is rich in inorganic cations. In contrast to the cation exchange membrane, only carbon and fluoride elements were found on both sides of the anion exchange membranes.

### 3.5. Membrane Fouling Speculation

Based on the comparison between the pristine and fouled membranes as well as the characterization data, the fouling mechanism of ion exchange membranes in treating tobacco extract solutions was proposed. For the anion exchange membranes, the fouling was possibly caused by the charged organic anions. This is because there are no metal ions precipitated on the surface of the anion exchange membranes as confirmed by the SEM-EDS analysis. It is known that most organic anions are negatively charged, these anions are migrating during the presence of the current field and will be disposed or adsorbed on the surface of membranes. Therefore, a gel layer is formed on the dilute side of the anion exchange after fouling [34]. Considering that the functional groups of the anion exchange membrane used for the experiments are the quaternary ammonium groups, the formation of negatively charged gel layers on the anion exchange membrane will accelerate the water dissociation on the membrane surface [35,36,37]. Notably, the increase of ion exchange capacity in fouled membranes also supports the organic fouling on the anion exchange membrane. Unfortunately, because the composition of tobacco extract is very complex, the possible fouling agents are difficult to be identified. However, considering that the tobacco extract is also one kind of plant juice, it is speculated that the possible fouling agents are the large-sized carboxylic acids such as humic acids, polyphenols, negative peptide charges (COO^−^), and amino acids [38,39,40]. For the cation exchange membrane, the fouling was mainly caused by minerals. This is confirmed by the decrease of ion exchange capacity for the fouled cation exchange membranes. Considering a large number of heavy metal ions in the tobacco extract solution, these metal ions shall be easily combined with the functional groups of cation exchange membranes. As a consequence, “membrane poisoning” will occur. The fouling by heavy metal ions on the surface cation exchange membranes is also supported by the SEM-EDS spectra. In the desalination process, the LCD of the tobacco extract solution will decrease with the desalting of the electrolyte solution. Water splitting will have happened on the surface of ion exchange membranes when the current exceeds the limiting current density. As discussed above, the limiting current density is around 8.3 mA/cm^2^ when the initial tobacco extract content is 7 wt.%, but the applied current is 3A (or current density is ~15 mA/cm^2^). The LCD was possibly reached for such a high demineralization rate (90%) in the ED experiments. In this case, the dissociated OH^-^ ions of water will combine with Ca^2+^ and Mg^2+^ ions and will precipitate on the surface of cation exchange membranes. The substantial inorganic scaling fouling in the cation exchange membrane is consistent with numerous studies [41,42,43,44]. Cifuentes-Araya and co-authors [42] found that divalent cations precipitation with OH^-^ ions produced by significant water splitting was reinforced by an important OH^-^ leakage through CEM that highly alkalinized the membrane interfaces, and also counteracted the demineralization. In a previous study [43], the authors systematically investigated the fouling nature evolutions along with different ED trials, and they found that amorphous calcium carbonate appeared on cation exchange membrane due to intense water-splitting reaction. Dufton and co-authors [44] have studied the fouling of ED for acid via deacidification, and they found the calcite and brucite were identified on the cation exchange membrane. The high solution alkalinity by the water dissociation was the reason for the fouling. All these studies have proven that the minerals such as Ca^2+^ and Mg^2+^ are the main elements for fouling on the cation exchange membrane.

### 3.6. Membrane Fouling Cleaning Strategies

To further mitigate the fouling of the membranes, two kinds of washing were conducted. One is an alkaline wash with 0.1 M NaOH for 15 min, the other one is an acid wash with 0.1 M HCl for 10 min. Acid and base washing is the most common methods for membrane cleaning strategies [4,33,45]. Figure 5 shows the conductivity and current evolution curves of the desalination process with different membrane washing strategies. It should be noticed that the membrane fouling experiments were finalized with 1200 mL solutions for the first batch run and then 400 mL solution for the other batch runs. It can be seen that without washing, the experimental time for a batch experiment in Run 3 is extended compared with that in Run 2, indicating the fouling of the membranes. Due to the fact that most organic matters have good solubility in alkaline circumstances, alkaline washing was conducted at first. However, only alkaline washing has a marginal impact on the mitigation of membrane fouling. As shown in Figure 5, the current of ED with only alkaline washing is even lower compared to that without any washing. Therefore, a combination of alkaline and acid washing was performed. It can be seen that with alkaline and acid washing, the membrane fouling was substantially mitigated. The experiment period was decreased for alkaline and acid washing mode compared to the no washing mode. The current evolution curve in Figure 6 indicated that the current increases in the alkaline and acid washing mode compared to no wash or alkaline washing mode. However, comparing Runs 5 and 6, it can be seen that even though the membrane fouling is mitigated with alkaline and acid washing, the fouling was still aggravated. The chemical washing can only mitigate but cannot avoid the fouling.

To further mitigate the membrane fouling, we developed a special cleaning agent based on the typical composition of tobacco extract. The composition of the cleaning agent is listed in Table 3. The cleaning agent is diluted to a 2 wt% concentration prior to practical use. Figure 6 indicates the desalination rate curves for ten consecutive ED experiments via this special cleaning agent. The special agent is applied after each batch experiment of ED. The desalination performances were slightly declined during the consecutive experiments, but the desalination rates became stable after the 6th experiment. The membrane fouling can be divided into reversible fouling and irreversible fouling. The reversible fouling can be recovered by chemical cleaning, but the irreversible fouling cannot be recovered. Nevertheless, this special cleaning agent containing a surfactant and metal-chelator is very effective for treating the tobacco extract. To increase the viability of electrodialytic desalting of tobacco extract, an elaborate pretreatment [27] and other mitigation strategies such as reversal ED or pulsed electric field [46,47,48,49] need to be further developed.

## 4. Conclusions

In this study, the fouling potential of the membranes was analyzed by comparing the properties for the pristine and fouled ion exchange membranes. The physical and chemical properties of the ion exchange membrane were analyzed in terms of electrical resistance, water content, and ion exchange capacity, as well as FT-IR spectra and SEM-EDS characterizations. The area resistance of the fouled anion and cation exchange membranes were increased 2–3 times and 5 times, respectively, compared to that of the pristine membranes. There is a slight decrease in water uptake in the fouled anion exchange membranes compared with fresh anion exchange membranes, but water uptake was obviously not changed for the cation exchange membranes. Large amounts of precipitates were observed on both sides of the anion exchange membranes. The calcium, magnesium, and sodium were found on the surface of cation exchange membranes, but only carbon and fluoride elements were observed on the surface of anion exchange membranes. The results showed that the membranes fouling is caused by two different mechanisms. For the anion exchange membranes, the fouling is mainly caused by the charged organic anions. For the cation exchange membrane, the fouling is caused by minerals ions such as Ca^2+^ and Mg^2+^. These metal ions reacted with OH^-^ ions produced through water dissociation and precipitated on the membrane surface. The chemical washing by alkaline and acid may mitigate, but cannot avoid the fouling of the ion exchange membranes. More efforts are still needed to further improve the long-term stability of ED for the desalting of tobacco extract solution.

## Figures and Tables

**Figure 1 membranes-10-00245-f001:**
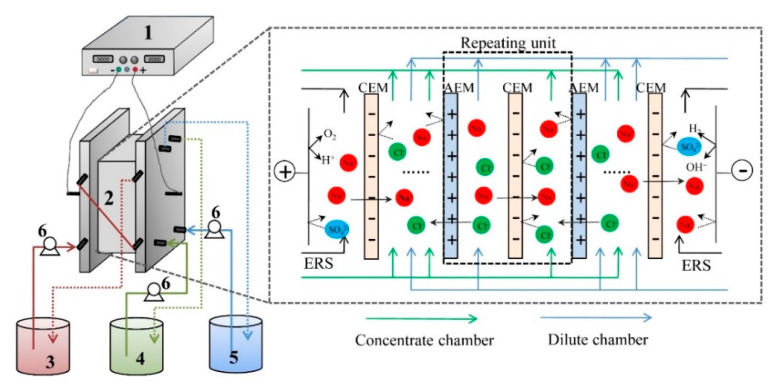
The main principle of electrodialysis for desalting of tobacco sheet extract. (**1**) Direct current supply; (**2**) Electrodialysis (ED) membrane stack; (**3**) Electrode rinse chamber; (**4**) Concentrate chamber; (**5**) Dilute chamber; and (**6**) Pump.

**Figure 2 membranes-10-00245-f002:**
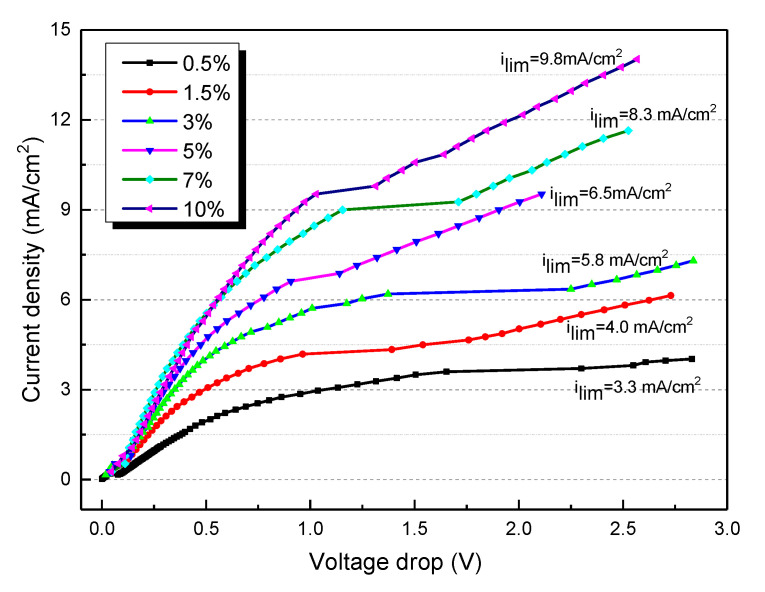
The current–voltage curves for different contents of tobacco extract.

**Figure 3 membranes-10-00245-f003:**
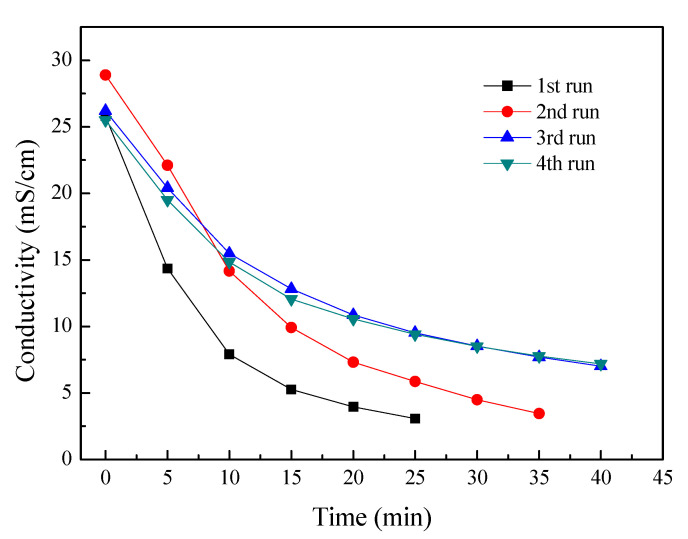
The conductivity curves for four consecutive ED experiments. Other experimental conditions: Feed in the dilute chamber, 1000 mL, 7% tobacco extract; Feed in the concentrate chamber, 1000 mL deionized water; Electrode rinse, 500 mL, 0.3 mol/L Na_2_SO_4_ aqueous solution; Operating mode: galvanostatic, 3 A.

**Figure 4 membranes-10-00245-f004:**
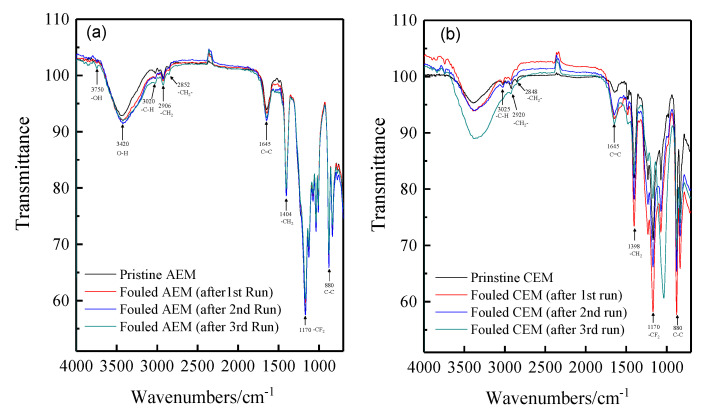
The FT-IR (ATR) spectra of the pristine and fouled anion and cation exchange membranes. (**a**) anion exchange membrane; (**b**) cation exchange membrane.

**Figure 5 membranes-10-00245-f005:**
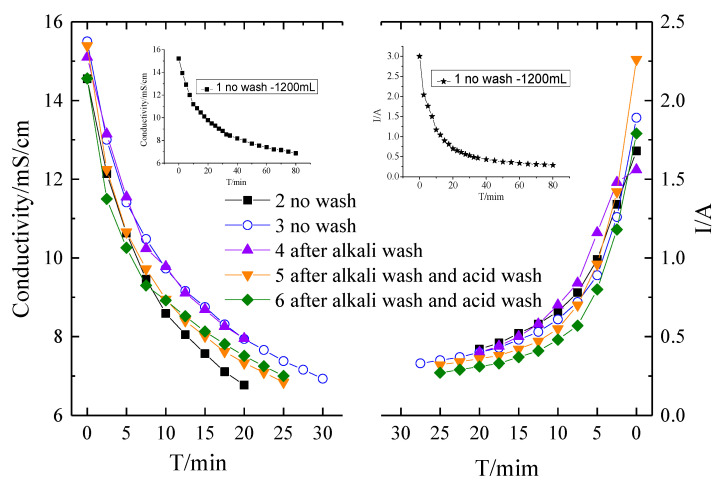
The conductivity and current evolution curves of the desalination process by different membrane washing strategies.

**Figure 6 membranes-10-00245-f006:**
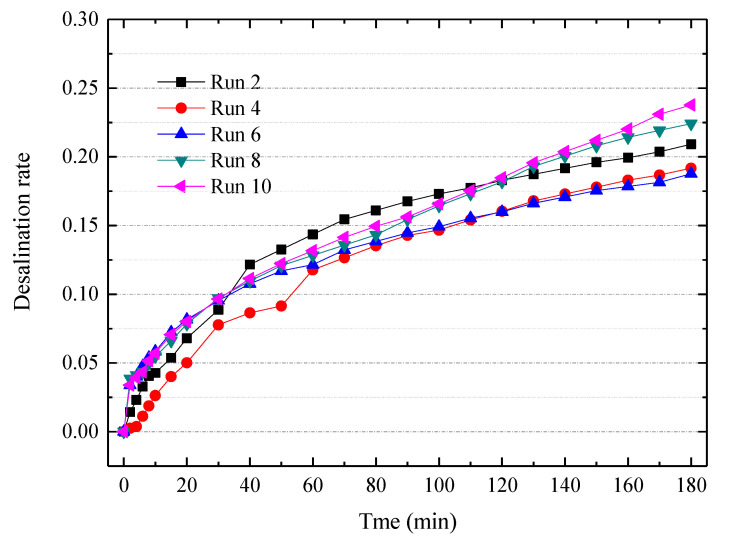
The desalination curve for ten consecutive ED experiments with a special cleaning agent. Other experimental conditions: Feed in the dilute chamber, 2000 mL 7% tobacco extract; Feed in the concentrate chamber, 1000 mL deionized water; Electrode rinse, 500 mL, 0.3 mol/L Na_2_SO_4_ aqueous solution; ED operating mode, potentiostatic mode 7 V.

**Table 1 membranes-10-00245-t001:** The main characteristics of tobacco extract.

Characteristics	Data
Total dissolved solids (wt.%)	7
Sugar (g/L)	17.31
Nicotine (g/L)	1.91
Nitrate (g/L)	672
Potassium (mg/L)	4255
Magnesium (mg/L)	654
Calcium (mg/L)	1217
Chloride (mg/L)	893
Sulfate (mg/L)	668
Malate (mg/L)	3537
Oxalate (mg/L)	35
Phosphate (mg/L)	501
Citrate (mg/L)	32
Tobacco extract solution pH	5.4
Tobacco extract conductivity (mS/cm)	22.4

**Table 2 membranes-10-00245-t002:** The main properties of the pristine and fouled membranes.

Membrane	IEC (meq/g)	Area Resistance (Ω cm^2^)	Water Uptake (%)	Transport Number (%)
CEM-pristine	0.8−1.0	2.5−3.5	40–45	>93
CEM-fouled	0.74	7.25	36.2	93
AEM-pristine	0.5−0.6	3.5−4.5	15–20	>93
AEM-fouled	1.11	16.04	16.7	90

**Table 3 membranes-10-00245-t003:** The composition of a special cleaning agent [50].

Compositions	Content
NaOH	30%
Tripolyphosphate	20%
Sodium dodecyl benzene sulfonate (SDS)	15%
Na_2_CO_3_	15%
NaCl	15%
EDTA	5%

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
