# Peer review of "Electrodialytic Desalination of Tobacco Sheet Extract: Membrane Fouling Mechanism and Mitigation Strategies"

_membranes, 2020, doi:10.3390/membranes10090245_

Round 1

Reviewer 1 Report

This study investigated the phenomenon of membrane fouling through physical and electrochemical characteristics and ED performance. The manuscript is well written, but some of contents in the introduction, method, and results are not clear and weak to convince readers. I would highly recommend the authors to revise the manuscript more carefully with regard to research significance, discussions, and English grammar in writing. Below are some comments the authors should reconsider and address in the manuscript to make this work publishable in the journal.

  1. The introduction lacks the statement of research significance and purpose. The author should enhance the introduction section and address it in the manuscript.
  2. In page 2, line 50, 64, nitrate symbol is not properly written.
  3. In page 3, line 122, it is confusing that tobacco extract is fed to dilute chamber and DI water to concentrate chamber, because tobacco extract is obviously much more concentrated than DI water. The author needs to clarify the reason for such arrangement.
  4. In page 4, line 138, an abbreviation, IEC was never explained in the content prior to this line. 
  5. In page 4, section 2.3 Characterization of ion exchange membranes, there is no mention of cation exchange membrane characterization method. It should be included in the context as well.
  6. In section 2.4 Calculation, the reviewer highly recommend to change the title of this section to "desalination rate". Otherwise, this sub-section can simply be merged into prior section. The authors should explain elaborate why R was calculated during membrane cleaning process. 
  7. In page 6, line 201, there is no mention of experimental period for 2nd batch of experiment. The author needs to include it in the context.
  8. In page 7, line 224-228, it is not very convincing statement that IEC has enhanced owing to adsorption of organic anions. To reviewer's point of view, the capacity of exchanging ion cannot simply be improved by adsorption of certain anions. The reviewer suggest to add more convincing reason (with references) to make such statement. 
  9. The thicknesses of investigated ion exchange membranes should also be noted in the context since it also influences the performance characteristics of the membrane to high extent. 
  10. In page 11, line 341-342, it is unclear of what it means by experimental period. The author should explain what it means by having extended period when had no washing. 
  11. In section 3.6 membrane fouling cleaning strategies, the author should explain why acid-washing alone is not conducted. The tests were only performed with base alone or combination of base and acid. 
  12. In page 12, line 360, 10 runs were tested for ED performance. However, it is not clear when exactly the special agent was applied. Is it after certain number of runs or before any runs? 
  13. The special agent for membrane cleaning was said to be developed in the context. To reviewer's opinion, at least some references should be added with some explanation to support such development.  

Author Response

Reviewer #1’s Comment: This study investigated the phenomenon of membrane fouling through physical and electrochemical characteristics and ED performance. The manuscript is well written, but some of contents in the introduction, method, and results are not clear and weak to convince readers. I would highly recommend the authors to revise the manuscript more carefully with regard to research significance, discussions, and English grammar in writing. Below are some comments the authors should reconsider and address in the manuscript to make this work publishable in the journal.

Response. We deeply appreciate the reviewer’s time and efforts in evaluating this manuscript. The reviewer’s constructive comments and suggestions help us improve the quality of the manuscript.

Q1. The introduction lacks the statement of research significance and purpose. The author should enhance the introduction section and address it in the manuscript.

A1. Thanks for the comments. There are numerous studies investigated about the fouling of membrane during electrodialysis process; but there are seldom studies about fouling of ion exchange membranes for treating tobacco sheet extract from industrial in a practical production line. Here, we developed readily adoptable methods to monitor the fouling of membranes, and elucidated the possible mechanisms for membrane fouling by tiny characterizations and by measuring the evolutions of the main properties of membranes during the ED experiments, and finally proposed an effective method to clean the fouled membranes with improved performances. This kind of the investigation of fouling in the ion exchange membranes can be extended to other kinds plant extracts. The statement about the novelty has been added in the revised manuscript (Page 2-3).

Q2. In page 2, line 50, 64, nitrate symbol is not properly written.

A2. Thanks for your conscientious, the wrong symbol of nitrate has been corrected as NO3-.

Q3. In page 3, line 122, it is confusing that tobacco extract is fed to dilute chamber and DI water to concentrate chamber, because tobacco extract is obviously much more concentrated than DI water. The author needs to clarify the reason for such arrangement.

A3. ED in this study is to selective remove the undesired ionic components such as Cl- , NO3-, we don’t want to introduce extra ions in the tobacco extract, so DI water was used as feed in the concentrate chamber, and tobacco extract was used as the feed in the dilute stream. During the ED experiments, the Cl-, NO3- ions are easily to be transported from the dilute chamber into the concentrate chamber, the conductivity in the concentrate chamber will increase very quickly. Even though the tobacco extract is much more concentrated than DI water, the undesired Cl-, NO3- ions will be removed in the presence of a direct current field.

Q4. In page 4, line 138, an abbreviation, IEC was never explained in the content prior to this line. 

A4. Thanks for the kind suggestion. Now IEC is explained when it is first time appeared in Section 2.3.

Q5. In page 4, section 2.3 Characterization of ion exchange membranes, there is no mention of cation exchange membrane characterization method. It should be included in the context as well.

A5. Thanks for the reviewer’s kind notice. To determine the ion exchange capacity of a cation exchange membrane, the membrane in H+ form was equilibrated in 1 mol/L aqueous NaCl solution for 24h to release the H+ ions, and then titration with standardized base using phenolphthalein as an acid-base indicator. This information has added in the revised manuscript.

Q6. In section 2.4 Calculation, the reviewer highly recommend to change the title of this section to "desalination rate". Otherwise, this sub-section can simply be merged into prior section. The authors should explain elaborate why R was calculated during membrane cleaning process. 

A6. According to the reviewer’s kind suggestion, the title of section 2.4 has been revised to “Desalination rate”. It should be noticed that it is easy to judge the effective of different cleaning methods by monitoring the changes in the desalination rate in consecutive batch operations. This information has been added in the revised manuscript.

Q7. In page 6, line 201, there is no mention of experimental period for 2nd batch of experiment. The author needs to include it in the context.

A7. Thanks for the useful comments. The related description has added in the revised manuscript “one batch experiment time increases to 35 mins for the 2nd run; and the experimental time further increases to 40 min at the 3rd and 4th batch of experiments” (Page 6).

Q8. In page 7, line 224-228, it is not very convincing statement that IEC has enhanced owing to adsorption of organic anions. To reviewer's point of view, the capacity of exchanging ion cannot simply be improved by adsorption of certain anions. The reviewer suggest to add more convincing reason (with references) to make such statement. 

A8. Thanks for the reviewer’s constructive comments. To address the reviewer’s concerns, we have revised the explanation for the increase of IEC of fouled anion exchange membrane compared with the pristine membrane. In this experiment, we found that the ion exchange capacity of the fouled anion exchange membrane was even higher than the pristine membrane. A possibly cause is the adsorption of organic matters such as protonated nicotine on the surface of anion exchange membrane. The nicotine molecules are easily to be protonated (as shown in the following scheme) into negative charged molecules [1], which then further deposit on the membrane surface due to the steric effect and the electric interaction. The revised explanation has been added in the revised manuscript. (Page 7)

Scheme 1. The molecule structure and protonated status of nicotine

[1] Clayton, P., Vas, C., and McAdam, K. (2014), “Use of chiroptical spectroscopy to determine the ionisation status of (S)-nicotine in electronic cigarette formulations”., ST 49, Coresta Congress, Québec City, Canada.

Q9. The thicknesses of investigated ion exchange membranes should also be noted in the context since it also influences the performance characteristics of the membrane to high extent. 

A9. Thanks for the reviewer’s comment. In our experiment, there is no significant change for the thickness of the membrane between the pristine and the fouled ion exchange membrane.

Q10. In page 11, line 341-342, it is unclear of what it means by experimental period. The author should explain what it means by having extended period when had no washing. 

A10. We have corrected the ambiguity description as “It can be seen that without washing, the experimental time for a batch experiment in Run 3 is extended compared with that in Run 2, indicating the fouling membranes”.

Q11. In section 3.6 membrane fouling cleaning strategies, the author should explain why acid-washing alone is not conducted. The tests were only performed with base alone or combination of base and acid. 

A11. In this study, the acid-washing is not independent investigated. According to the change of the main properties of the pristine and fouled membranes in Table 2, we speculated the fouling of anion exchange membrane is possibly caused by the organic anions; while the fouling of cation exchange membrane is highly related to the minerals. In this case, to remove the anionic and mineral foulants, both the acid-washing and alkali-washing are required. But if the acid wash is conducted at first and then followed by the alkali wash, it maybe causes secondary fouling due to the formation of metal hydroxide. Therefore, the base wash is conducted at first and then followed by the acid wash.

Q12. In page 12, line 360, 10 runs were tested for ED performance. However, it is not clear when exactly the special agent was applied. Is it after certain number of runs or before any runs? 

A12. The special agent is applied after each batch experiment of ED. This information has added in the revised manuscript.

Q13. The special agent for membrane cleaning was said to be developed in the context. To reviewer's opinion, at least some references should be added with some explanation to support such development.  

A13. Thanks for the comments. The special agent was developed based on the literature [1]; the related reference has been added according to the reviewer’s kind suggestion.

[1] Zhu, A.M.; Teng, H.K.; Zheng, S.Z.; Chen, J.; Zhang, D.M.; Zhang, L. A cleaning agent and its manufacturing methods for ion exchange membrane used for electrodialysis. 2013, China Patent., CN. 103463991A.

Reviewer 2 Report

This manuscript presents the fouling potential of the cation exchange and anion exchange membranes using ED method, and suggested the membrane fouling mechanisms and a cleaning method. Overall it is well-expressed. But the following concerns were raised:

1). The introduction part should be improved to answer the following questions:

a). Is there any former study on the mechanism of membrane fouling?

b). Is there any other method/strategy to clean fouling except the acid and base washing method mentioned in the manuscript?

c). What is the improvement or significant contribution of this study to this field comparing to previous studies?

2). This work used FT-IR method to analyze the components left on membrane. Is there any other more reliable method available to study the residue composition?

3). Some chemical reactions happened in the ED process based on the context, which should be expressed in the manuscript.

Author Response

This manuscript presents the fouling potential of the cation exchange and anion exchange membranes using ED method, and suggested the membrane fouling mechanisms and a cleaning method. Overall it is well-expressed. But the following concerns were raised:

Response. We deeply appreciate the reviewer’s expertise in evaluating our manuscript. Your constructive comments and suggestions help us improve the quality of the present work.

Q1). The introduction part should be improved to answer the following questions:

a). Is there any former study on the mechanism of membrane fouling?

b). Is there any other method/strategy to clean fouling except the acid and base washing method mentioned in the manuscript?

c). What is the improvement or significant contribution of this study to this field comparing to previous studies?

A1. Thanks for the constructive comments. The following description has been added in the introduction part in Page 2-3.

In fact, there are numerous studies investigated about the fouling of membrane during electrodialysis process. According to the review summarized by Bazinet [1], there are usually four kinds of fouling in the ion exchange membrane such as colloidal, organic, scaling and biofouling. To elucidate the evolution of fouling in the ion exchange membranes, various methods have been used including visualization the fouling by photo imaging, scanning electron microscopy (SEM) and atomic force microscopy (AFM), monitoring the evolution of the membrane characteristics such as membrane electrical resistance and conductivity, voltammetry and chronopotentiometry, transport number, zeta potential and contact angles, and analysis the fouling composition such as spectroscopy, chromatography and biofouling characterization.

To mitigate the fouling of the membranes, numerous strategies allowing the fouling prevention have been proposed such as by using cleaning agent, modifying the ion exchange membranes, introducing elaborated pretreatment, involving mechanical cleaning such as ultrasound, vibration and air sparge, and by changing regimes of ED treatment.

But there are rare studies on the fouling of ion exchange membranes for treating tobacco sheet extract from a practical production line. Here, we developed readily adoptable methods to monitor the fouling of membranes, and elucidated the possible mechanisms for membrane fouling by tiny characterizations and by measuring the evolutions of the main properties of membranes, and finally proposed an effective method to clean the fouled membranes with improved performances. This kind of the investigation of fouling in the ion exchange membranes can be extended to other kinds of plant extracts.

[1] Mikhaylin, S.; Bazinet, L. Fouling on ion-exchange membranes: Classification, characterization and strategies of prevention and control. Adv. Colloid Interface Sci. 2016, 229, 34-56.

2). This work used FT-IR method to analyze the components left on membrane. Is there any other more reliable method available to study the residue composition?

A2. In this study, we used FT-IR and SEM-EDS as the tiny characterization methods to analyze the components of foulants. Of course, some other methods such as visualization the fouling by photo imaging, analysis the nitrogen content by combustion of the membranes and by analysis the X-ray diffraction analysis (XRD) of the membranes. But due to complex of tobacco extract, these tiny characterizations were not convenient compared with FT-IR and SEM-EDS. In our case, the physical and chemical properties of the ion exchange membrane were analyzed in terms of electrical resistance, water content, and ion exchange capacity, as well as FT-IR spectra and SEM-EDS characterizations. The mechanisms for the membrane fouling were proposed and cleaning strategies for fouling prevention were also developed.

3). Some chemical reactions happened in the ED process based on the context, which should be expressed in the manuscript.

A3. Theoretically, the ED is a physical separation process and there are no chemical reactions happened except the electrode reactions. But there are ten repeating unit of membranes for one pair of electrodes, the electrode reactions are neglectable. Therefore, the chemical reactions were not considered in the experiments.

Reviewer 3 Report

This study reports an electrodialytic approach for treatment of tobacco sheet extract with emphasis on membrane fouling mechanism and mitigation strategies. However, authors already published almost similar works in several publications (Zhang, Z.H.; Ge, S.L.; Jiang, C.X.; Zhao, Y.; Wang, Y.M. Improving the smoking quality of papermaking tobacco sheet extract by using electrodialysis. Membr. Water. Treat. 2014, 5, 31-40.;

Ge, S. L.; Li W.; Zhang, Z.; Li, C.R.; Wang, Y.M. Desalting tobacco extract using electrodialysis. Membr. Water Treat. 2016, 7, 341-353.

Li, C.R.; Ge, S.L.; Li, W.; Zhang, Z.; She, S.K.; Huang, L.; Wang, Y.M. Desalting of papermaking tobacco sheet extract using selective electrodialysis. Membr. Water. Treat. 2017, 8, 381-393.).

Thus, this work is not recommended for publication in the Membranes due to the lack of novelty.

Author Response

This study reports an electrodialytic approach for treatment of tobacco sheet extract with emphasis on membrane fouling mechanism and mitigation strategies. However, authors already published almost similar works in several publications (Zhang, Z.H.; Ge, S.L.; Jiang, C.X.; Zhao, Y.; Wang, Y.M. Improving the smoking quality of papermaking tobacco sheet extract by using electrodialysis. Membr. Water. Treat. 2014, 5, 31-40.;

Ge, S. L.; Li W.; Zhang, Z.; Li, C.R.; Wang, Y.M. Desalting tobacco extract using electrodialysis. Membr. Water Treat. 2016, 7, 341-353.

Li, C.R.; Ge, S.L.; Li, W.; Zhang, Z.; She, S.K.; Huang, L.; Wang, Y.M. Desalting of papermaking tobacco sheet extract using selective electrodialysis. Membr. Water. Treat. 2017, 8, 381-393.).

Thus, this work is not recommended for publication in the Membranes due to the lack of novelty.

Response. We deeply appreciate the reviewer’s time and efforts in evaluating our manuscript. In fact, the present work is focused on the fouling mechanism and fouling prevention strategies, that is quite different from our previous publications. To address the reviewer’s concerns on the novelty of this study, we have strengthened the research significance and purpose of this work in the revised manuscript. Even though there are numerous studies investigated about the fouling of membrane during electrodialysis process, there are rare studies on the fouling of ion exchange membranes for treating tobacco sheet extract from a practical production line. Here, we developed readily adoptable methods to monitor the fouling of membranes, and elucidated the possible mechanisms for membrane fouling by tiny characterizations and by measuring the evolutions of the main properties of membranes, and finally proposed an effective method to clean the fouled membranes with improved performances. This kind of the investigation of fouling in the ion exchange membranes can be extended to other kinds of plant extracts.

Reviewer 4 Report

The study, described the authors, on using ED membranes for the desalination of tobacco sheet extract is an interesting one. The results in this paper would be potentially useful in guiding future work in this area. Having said that, the article does not describe any mechanism/hypotheses in details. 

  1. With respect to the discussion related to Table 2, while the authors have noted some interesting and a few counter-intuitive observations, they do not present enough reasoning to explain such observations (e.g. increase in area resistance of fouled vs. pristine membranes, changes in water uptake & transport number in fouled membranes). Some explanations are needed. 
  2. In line 232, I believe it should be "Table 2" instead of "Table 1". 
  3. The main focus of this paper is on fouling and mitigation strategies; it would be better if (i) the authors quantified the extent of irreversible and reversible fouling, (ii) a control experiment on a "pristine" membrane with the cleaning agent would be good to determine if that agent has any adverse effect on the membrane properties. 
  4. In fig. 5, the authors make some important conclusions based on SEM EDS. This is not a very robust technique and I believe it would be better to put this data in supplementary info. 
  5. Lastly, the writing of this manuscript can be improved; it is a little difficult to follow some portions of the paper because of the writing. 

Author Response

The study, described the authors, on using ED membranes for the desalination of tobacco sheet extract is an interesting one. The results in this paper would be potentially useful in guiding future work in this area. Having said that, the article does not describe any mechanism/hypotheses in details. 

Response. We deeply appreciate the reviewer’s expertise in evaluating our manuscript. Your constructive comments and suggestions help us improve the quality of the present work.

Q1. With respect to the discussion related to Table 2, while the authors have noted some interesting and a few counter-intuitive observations, they do not present enough reasoning to explain such observations (e.g. increase in area resistance of fouled vs. pristine membranes, changes in water uptake & transport number in fouled membranes). Some explanations are needed. 

A1. Thanks for the reviewer’s constructive comments. According to the reviewer’s comments, we have revised the explanation on the increase of IEC of fouled membrane compared with the pristine membranes. A possibly cause is the adsorption of organic matters such as protonated nicotine on the surface of anion exchange membrane. The nicotine molecules are easily to be protonated (as shown in the following scheme) into negative charged molecules, which then further deposit on the membrane surface due to the steric effect and the electric interaction.

Scheme 1. The molecule structure and protonated status of nicotine molecules

The explanation for the increase of membranes resistance were also added in the revised manuscript as “The fouling of ion exchange membranes leads to the increase of membrane resistance, which are possibly related to two causes. One cause is due to the formation of surface fouling layer and the other cause is due to the deposition of foulants on the functional groups of ion exchange membranes”.

Q2. In line 232, I believe it should be "Table 2" instead of "Table 1". 

A2. Corrected as suggested.

Q3. The main focus of this paper is on fouling and mitigation strategies; it would be better if (i) the authors quantified the extent of irreversible and reversible fouling, (ii) a control experiment on a "pristine" membrane with the cleaning agent would be good to determine if that agent has any adverse effect on the membrane properties. 

A3. Thanks for the reviewer’s constructive comments. It is very desirable if we can quantify the extent of irreversible and reversible membrane fouling. But due to complex of the composition of tobacco extract in the feed, it is very difficult to identify the detailed fouling matters. Therefore, it is difficult to quantify the irreversible and reversible fouling if the fouling matters are not identified. We have soaked the membrane in the cleaning agent and found there are no obvious changes in the membrane resistance and ion exchange capacity, so it is speculated that the cleaning agent has no significant adverse effect on the membrane properties. In fact, the cleaning agent was developed based on the literature, and the reference was added in the revised manuscript according to one reviewer’s comment.      

Q4. In fig. 5, the authors make some important conclusions based on SEM EDS. This is not a very robust technique and I believe it would be better to put this data in supplementary info. 

A4. Thanks for the reviewer’s kind suggestion. The fig. 5 was provided as supplementary info.

Q5. Lastly, the writing of this manuscript can be improved; it is a little difficult to follow some portions of the paper because of the writing. 

A5. Thanks for the comments. We have proofread the manuscript carefully to correct the typo errors and have asked several colleagues of skilled English speaker to improve the language.

Reviewer 5 Report

I like this paper. It is quite interesting, although some previous papers of Authors have similar content.

I do appreciate that research have been conducted for the medium that contains thousands of matters. The chapter 3. 5 Membrane fouling speculation is in my opinion the strongest element in the paper. One can find there very inspiring information.

I find spelling mistaken the line 147. There is (Wu) while in the equation there is WR.

Author Response

I like this paper. It is quite interesting, although some previous papers of Authors have similar content.

I do appreciate that research have been conducted for the medium that contains thousands of matters. The chapter 3. 5 Membrane fouling speculation is in my opinion the strongest element in the paper. One can find there very inspiring information.

I find spelling mistaken the line 147. There is (Wu) while in the equation there is WR.

Response. We deeply appreciate the reviewer’s positive comments on this manuscript. The spelling mistake of water uptake in line 147 has been corrected for the reviewer’s kindly reminding us.

Round 2

Reviewer 1 Report

The authors have satisfactorily addressed and reflected all the comments raised by reviewer. The revised version of manuscript looks publishable to the journal.

Author Response

Thanks again for the reviewer's time and expertise in evaluating this manuscript. 

Reviewer 2 Report

Small grammar errors were found in the current context, which should be fixed. 

Author Response

Thanks for the reviewer's time and expertise in evaluating this manuscript. We have proofread the manuscript again to correct the typo errors.